# Sarcopenia as an independent prognostic marker in liposarcoma: A longitudinal analysis of body composition and survival

Julian Kylies[1,2]*, Matthias Priemel[1,2], Anna Dupree[2,3], Karl-Heinz Frosch[1,4], Tobias M. Ballhause[1,2]

**1** Department of Trauma and Orthopedic Surgery, University Medical Center Hamburg-Eppendorf, Hamburg, Germany, **2** Hubertus Wald University Cancer Center Hamburg, University Medical Center Hamburg-Eppendorf, Hamburg, Germany, **3** Department of General, Visceral and Thoracic Surgery, University Medical Center Hamburg-Eppendorf, Hamburg, Germany, **4** Department of Trauma Surgery, Orthopaedics and Sports Traumatology, BG Klinikum Hamburg, Hamburg, Germany

* j.kylies@uke.de

## Abstract

### Background

Sarcopenia is increasingly recognized as an important prognostic factor in oncology; however, its clinical relevance in liposarcoma remains insufficiently defined. This study aimed to evaluate longitudinal changes in CT-derived body composition parameters in liposarcoma patients, to assess the influence of tumor grade, recurrence, and treatment modalities on these parameters and to determine the association of baseline sarcopenia and progressive muscle loss with overall survival and functional status.

### Methods

In a retrospective, single center study between 2010 and 2024, 64 patients were analyzed. All patients underwent surgical tumor resection of a histologically confirmed liposarcoma. Included were patients with two consecutive CT scans. The following morphometric parameters were measured on CT axial images at the height of lumbar vertebral 3: Skeletal muscle index (SMI), paraspinal muscle index (PSMI), psoas muscle index (PMI), skeletal muscle density (SMD), and visceral adipose tissue (VAT). Standardized Hounsfield unit thresholds were used for the assessment. Additionally, the influence of tumor grade, recurrence, and treatment modalities on body composition was assessed. A Kaplan Meier survival analysis was performed using data from the residents´ registration office. Survival was further analyzed by Cox regression using uni- and multivariate models. Metric data was compared using student´s t-test.

**Data availability statement:** All relevant data are within the paper and its Supporting information files.

**Funding:** The author(s) received no specific funding for this work.

**Competing interests:** The authors have declared that no competing interests exist.

## Results

Significant reductions in SMI, PSMI, PMI, and VAT were observed over the disease course, particularly among patients with high-grade tumors, chemotherapy, or local tumor recurrence. Baseline sarcopenia and a progressive SMI loss were independently associated with reduced overall survival. In multivariate analysis, baseline sarcopenia (HR: 2.331, $p = 0.007$) and a $\geq 15\%$ SMI decline (HR: 2.601, $p = 0.006$) remained significant predictors of mortality. Both markers did not correlate with changes in Eastern Cooperative Oncology Group (ECOG) performance status.

## Conclusion

CT-morphometric parameters deteriorate substantially during the disease course of liposarcoma patients and serve as independent predictors of survival. These findings support the integration of CT-based body composition analysis into routine oncologic assessment and highlight its potential role in identifying high-risk patients for early supportive intervention.

## Introduction

Liposarcoma is the most prevalent subtype of soft tissue sarcomas in adults, characterized by diverse histological variants and a variable clinical course [1–4]. Despite advancements in surgical techniques and multimodal therapies, prognostic assessment remains challenging, especially in intermediate- to high-grade liposarcomas [5–7].

Sarcopenia, defined as the pathological loss of skeletal muscle mass and function, has emerged as a significant prognostic factor in oncology. Initially described in the context of aging, sarcopenia is now increasingly recognized as a marker of disease-related frailty in cancer patients [8–10]. Its presence has been linked to a spectrum of adverse clinical outcomes, including increased chemotherapy toxicity, higher rates of postoperative complications, and diminished tolerance to multimodal treatment [8–12]. Multiple studies have demonstrated that reduced skeletal muscle mass is associated with reduced survival outcomes across various malignancies, including colorectal, and lung cancer [13–15]. Recently, CT-morphometric assessment of sarcopenia has been increasingly utilized in this context [16]. Using routine cross-sectional CT scans, which are often already performed for staging and surveillance purposes, CT-based morphometric analysis allows for the assessment of skeletal muscle indices measured in standardized Hounsfield Units.

While CT-derived sarcopenia metrics have been extensively studied in other malignancies, their relevance in liposarcoma, remains underexplored. In two recent studies, the prognostic relevance of CT-based body composition analysis in patients with high-risk soft tissue sarcomas undergoing multimodal therapy was analyzed [13–15]. These studies suggest that unfavorable changes in skeletal muscle and fat

compartments are associated with poorer overall survival [17,18]. But heterogeneous sarcoma types were mixed in the analysis, limiting the ability to draw specific conclusions [19,20].

Therefore, the aim of this study was to investigate body composition dynamics over the disease course in a well-characterized cohort of liposarcoma patients undergoing surgical resection. Specifically, we (1) evaluated longitudinal changes in CT-derived morphometric parameters (SMI, PSMI, PMI, SMD, and VAT), (2) examined the prognostic value of baseline sarcopenia and progressive skeletal muscle loss on overall survival and functional outcomes, and (3) analyzed the impact of tumor grade, recurrence, and treatment modalities on these body composition trajectories. By focusing exclusively on liposarcoma, this study provides entity-specific insights into the prognostic and clinical relevance of sarcopenia in soft-tissue sarcoma.

## Materials and methods

This retrospective observational study was approved by the Ethics Committee of the Hamburg Medical Association (ethics ID: 2025–300576-WF) and conducted in accordance with the principles of the Declaration of Helsinki. In view of the fact that the patient data that are the subject of the study can no longer be attributed to a human being, the study does not constitute a "research project involving human beings" as defined in Section 9 (2) of the Hamburg Chamber Act for the Medical Professions and also does not fall within the scope of the research projects requiring consultation pursuant to Section 15 (1) of the Professional Code of Conduct for Hamburg Physicians. Therefore, the requirement for informed consent was waived.

A total of 64 patients with histologically confirmed liposarcoma who underwent surgical resection between 2010 and 2024 were included. The cohort encompassed liposarcomas which were histologically graded as G1, G2 or G3 according to the 2020 WHO Classification of soft tissue and bone tumors [21]. Thus, atypical lipomatous tumors, formally known as "well-differentiated liposarcomas" were not included. Eligibility criteria required the availability of at least two CT scans per patient: an initial scan performed prior to surgical intervention and a second scan acquired roughly 12 months later. These defined the two time points, tCT1 and tCT2, used for morphometric analysis as well as clinical data acquisition. An overview of the patient inclusion process is provided in **Fig 1A**.

Patients were included if baseline imaging covered the L3 vertebral level with sufficient quality for morphometric segmentation and if comprehensive clinical data, such as Eastern Cooperative Oncology Group (ECOG) performance status, resection margin, TNM staging, chemotherapy, and radiotherapy, were available at both imaging time points. Patients were excluded if they had fewer than two eligible CT scans, imaging artifacts (e.g., motion, metal implants), incomplete coverage of the lumbar vertebral 3 level, or a history of prior or concurrent malignancies that could interfere with morphometric interpretation. Additional exclusion criteria included prior spinal surgery, musculoskeletal diseases (e.g., muscular dystrophy), or incomplete clinical documentation at either time point.

CT-based morphometric analysis was performed using Fiji (ImageJ) software (Version 2.3.0/1.53q; Max Planck Institute of Molecular Cell Biology and Genetics, Dresden, Germany). Skeletal muscle cross-sectional areas were measured at the lumbar vertebral 3 using standard Hounsfield unit (HU) thresholds from −29 to +150 HU, as previously described [22,23]. Values were normalized to patient height (cm²/m²) to compute the SMI, PSMI and PMI. SMD, as a surrogate for muscle quality, was calculated as the mean HU value of paraspinal musculature. VAT was segmented using a range of −190 to −30 HU on the same axial slice (Fig 1B-E) [22,23].

All CT-based morphometric measurements were performed by a single trained observer using a semi-automated segmentation workflow to ensure consistency across scans. Standardized Hounsfield Unit thresholds and pre-defined region-of-interest boundaries at the L3 vertebral level were applied to minimize operator-related variability. Formal inter-rater reliability testing was not conducted in this retrospective dataset.

Clinical and laboratory parameters were retrieved from institutional electronic medical records and matched to the respective imaging time points (tCT1 and tCT2) for all patients. Clinical and CT-morphometric data were obtained as of March to April 1, 2025.

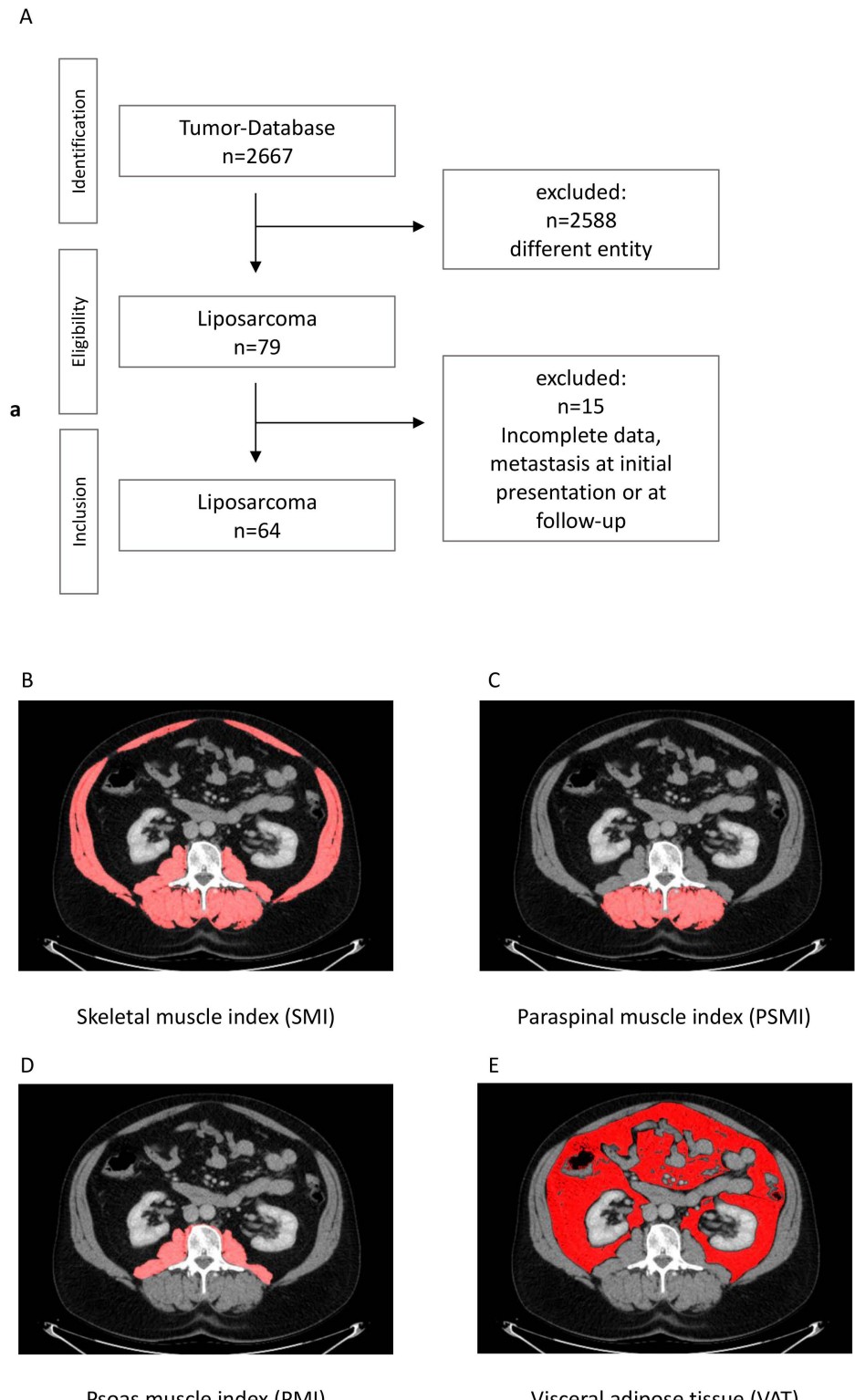

Skeletal muscle index (SMI)

Paraspinal muscle index (PSMI)

Psoas muscle index (PMI)

Visceral adipose tissue (VAT)

**Fig 1. (A)** Overview of the patient selection and study design. **(B–E)** Representative CT slices demonstrating morphometric segmentation of skeletal muscle and visceral adipose tissue at the lumbar 3 vertebral level. Cross-sectional muscle area was used to compute the skeletal muscle index (SMI), psoas muscle index (PMI), paraspinal muscle index (PSMI), skeletal muscle density (SMD), and visceral adipose tissue (VAT).

## Statistical analysis

All statistical analyses were conducted using IBM SPSS Statistics version 29 (IBM Corp., Armonk, NY, USA), while graphical plots were generated using GraphPad Prism version 10.2.2 (GraphPad Software, La Jolla, CA, USA). Differences in CT-based morphometric parameters between the two time points were assessed using the non-parametric Mann-Whitney U test. Differences between three or more groups were assessed using non-parametric Kruskal-Wallis test. Continuous variables are reported as mean ± standard deviation (SD) and p-value < 0.05 was considered statistically significant.

## Survival analysis

Survival data were obtained from residents' registration office as of April 1, 2025. The mean follow-up duration for survival analysis was 79.2 ± 48.2 months. For survival analyses, only patients between 2010 and 2020 were included to ensure a follow-up of at least 5 years. Kaplan-Meier survival curves were constructed to compare overall survival between sarcopenic and non-sarcopenic patients at baseline, as well as between patients who experienced a ≥ 15% vs. < 15% reduction in SMI during the disease course. Group differences were evaluated using the log-rank (Mantel-Cox) test. Receiver Operating Characteristic (ROC) curve analysis was conducted to identify the optimal threshold of SMI decline predictive of survival, with Youden's Index applied to determine the most discriminative cut-off. A 15% reduction in SMI emerged as the strongest predictor of decreased survival probability.

## Cox proportional hazards regression

To assess whether CT morphometric parameters independently predicted overall survival, a multivariate Cox proportional hazards model was developed. The model was adjusted for potential confounders including patient age, sex, ECOG performance status, tumor recurrence, and maximum tumor diameter. All patients included in the survival analysis had a baseline TNM classification of N0 and M0. Hazard ratios (HRs) with corresponding 95% confidence intervals (CIs) were calculated, and partial likelihood estimation was performed using the Efron method to resolve tied event times. A significance threshold of $p < 0.05$ was applied to all model covariates.

# Results

## Patient demographics and study design

A screening of our local tumor database between 2010 and 2024 was performed. Initially, 2667 sarcoma patients were identified. After excluding all non-liposarcoma entities, 79 patients with histologically confirmed liposarcoma were identified. Of these, 15 individuals were excluded due to insufficient imaging data, incomplete clinical documentation, or lack of follow-up CT scans, resulting in a final study cohort of 64 patients.

The cohort included 28 female patients, with a mean age of 60.0 ± 16.5 years at the time of diagnosis. Tumor grading distribution showed that 45 patients presented with high grade liposarcomas (G2-3), while 19 patients had low grade tumors (G1). A summary of the baseline demographic and clinical characteristics is provided in **Table 1**.

The average time between the two selected CT examinations (tCT1 and tCT2) was 12.3 ± 2.6 months, which allowed for the assessment of longitudinal changes in CT-derived body composition metrics. Due to the differences in absolute skeletal muscle and adipose tissue distribution between sexes, sex-stratified morphometric data are presented in **Table 2**. Relative changes in these parameters over time were analyzed across the entire cohort enabling uniform comparison and interpretation.

## Longitudinal decline in CT-morphometric sarcopenia and adiposity parameters in liposarcoma patients

Over the disease course, significant declines in CT-derived body composition parameters were observed across the entire liposarcoma cohort (both male and female).

**Table 1. Baseline demographic and clinical characteristics of the study cohort (n = 64), including age, sex, tumor grade, and treatment modalities.**

| Total (n) | 64 |
|---|---|
| Female | 28 (44%) |
| Male | 36 (56%) |
| **Mean Age (years)** | 60 ± 16.5 years |
| Female | 59.1 ± 17.1 years |
| Male | 61.3 ± 15.9 years |
| **Mean follow up (in months)** | 12.3 ± 2.6 months |
| **Treatment** | |
| Surgery only | 31 (48%) |
| Surgery/Chemotherapy | 24 (38%) |
| Surgery/Radiotherapy | 6 (9%) |
| Surgery/Chemotherapy/ Radiotherapy (excluded) | 3 (5%) |
| **Grading** | |
| G1 | 19 (29%) |
| G2 | 7 (11%) |
| G3 | 38 (60%) |
| **Local tumor recurrence** | 19 (29%) |
| **No Local tumor recurrence** | 45 (71%) |

**Table 2. Sex-stratified CT morphometric parameters at baseline (tCT1) and follow-up (tCT2), with corresponding p-values.**

| | tCT1 (Mean/SD) | tCT2 (Mean/SD) |
|---|---|---|
| **Skeletal Muscle Index (SMI) [cm²/m²]** | 47.5 ± 6.4 | 34.9 ± 9.1 |
| Male | 50.7 ± 6.5 | 36.1 ± 9.8 |
| Female | 44.3 ± 6.3 | 33.8 ± 8.6 |
| **Paraspinal Muscle Index (PSMI) [cm²/m²]** | 18.5 ± 2.0 | 10.7 ± 3.3 |
| Male | 19.1 ± 2.2 | 11.3 ± 3.5 |
| Female | 17.2 ± 1.7 | 9.9 ± 2.9 |
| **Psoas Muscle Index (PMI) [cm²/m²]** | 2.8 ± 0.6 | 1.1 ± 0.5 |
| Male | 2.8 ± 0.6 | 1.1 ± 0.5 |
| Female | 2.8 ± 0.7 | 1.1 ± 0.7 |
| **Skeletal Muscle Density (SMD)** | 45.1 ± 6.3 | 42.1 ± 8.4 |
| Male | 45.7 ± 6.3 | 43.4 ± 9.3 |
| Female | 44.6 ± 6.2 | 41.4 ± 7.8 |
| **Visceral Adipose Tissue (VAT) [cm²]** | 84.3 ± 9.9 | 65.1 ± 9.4 |
| Male | 90.3 ± 10.2 | 68.6 ± 8.9 |
| Female | 81.2 ± 9.8 | 62.3 ± 10.0 |

The SMI decreased markedly from 47.5 ± 6.4 cm²/m² at tCT1 to 34.9 ± 9.1 cm²/m² at tCT2 ($p < 0.001$, **Fig 2A**). Similarly, the PSMI declined significantly from 18.5 ± 2.0 cm²/m² to 10.7 ± 3.3 cm²/m² ($p < 0.001$, **Fig 2B**), and the PMI dropped from 2.8 ± 0.6 cm²/m² to 1.1 ± 0.5 cm²/m² ($p < 0.001$, **Fig 2C**). SMD also showed a statistically significant, though modest, reduction from 45.1 ± 6.3 HU to 42.1 ± 8.4 HU ($p = 0.03$, **Fig 2D**).

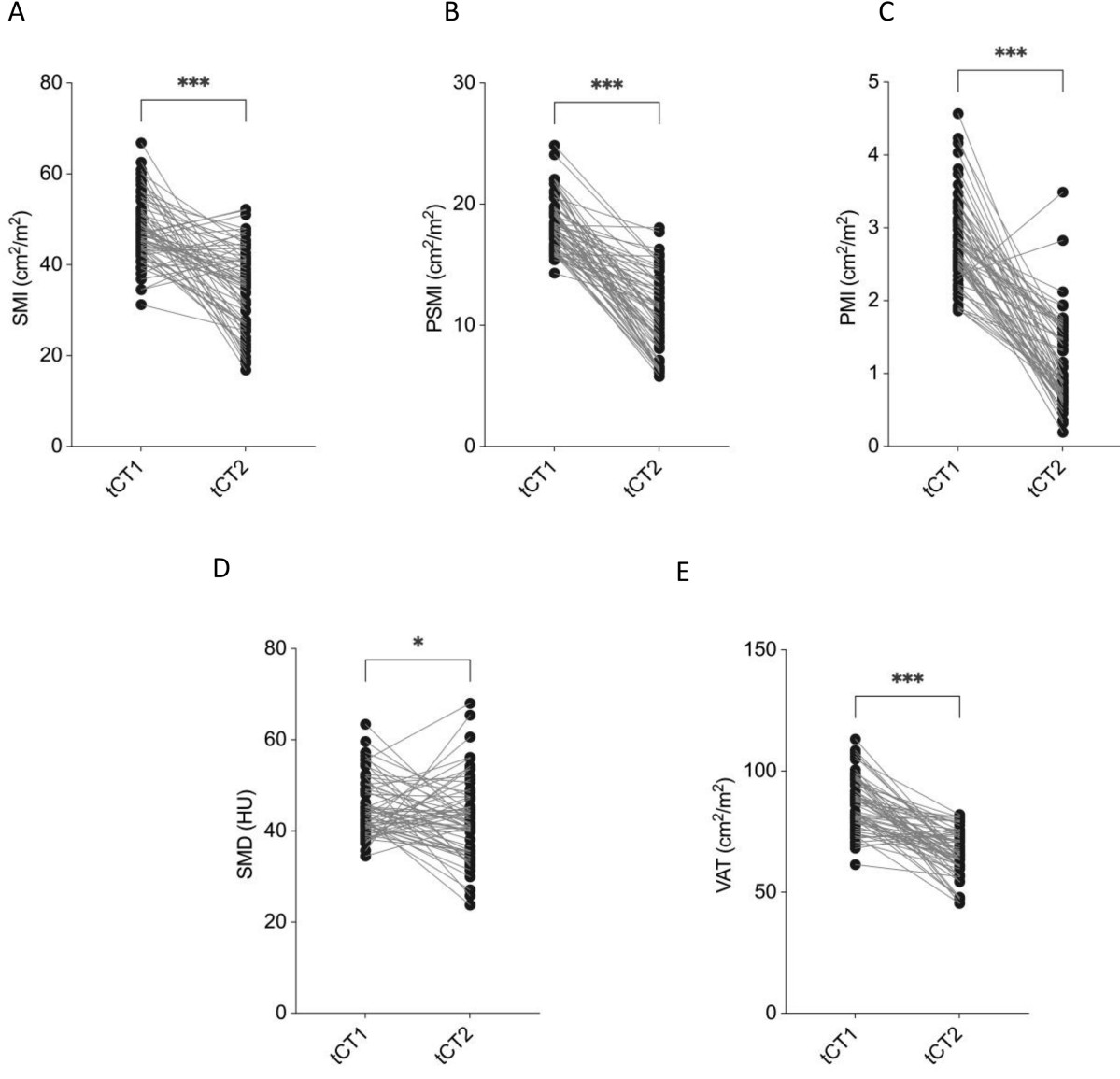

**Fig 2. Longitudinal changes in CT-derived morphometric parameters in liposarcoma patients.** Significant reductions were observed in Skeletal muscle index (SMI) **(A)**, Paraspinal muscle index (PSMI) **(B)**, Psoas muscle index (PMI) **(C)**, Skeletal muscle density (SMD) **(D)** and Visceral adipose tissue (VAT) **(E)** between tCT1 and tCT2. Non-parametric Mann-Whitney U test has been applied.

The VAT area decreased substantially from 84.3±9.9 cm² to 65.1±9.4 cm² (p<0.001, **Fig 2E**).

Collectively, these results highlight a consistent and significant loss of skeletal muscle mass, muscle density, and visceral fat across the study population during the disease course.

## Tumor grade-dependent differences in CT-morphometric parameter changes

To explore the influence of tumor grade on body composition changes, patients were stratified into low grade (G1) and high grade (G2-3) liposarcoma subgroups. Notable differences in the degree of morphometric decline were observed between these groups over the disease course.

To assess the impact of tumor grade on longitudinal changes in body composition, patients were stratified into low-grade (G1) and high-grade (G2–3) liposarcoma groups. Patients with high-grade tumors exhibited significantly greater reductions in multiple CT-morphometric parameters compared to those with low-grade disease.

The mean percentage decrease in SMI was markedly higher in the G2–3 group compared to G1 patients (−35.0 ± 19.9% vs. −7.5 ± 21.1%, p < 0.001; **Fig 3A**). Similarly, the decline in PSMI was significantly more pronounced among high-grade tumors (−47.8 ± 15.6%) than in low-grade tumors (−31.5 ± 17.1%, p < 0.001; **Fig 3B**). PMI also showed a significantly steeper decline in the G2–3 group (−65.8 ± 17.0%) compared to the G1 group (−42.5 ± 33.7%, p < 0.01; **Fig 3C**). SMD demonstrated a slight decrease in the high-grade group (−5.8 ± 20.9%) but remained stable in the low-grade group (0.1 ± 27.8%), with this difference not reaching statistical significance (p = n.s.; **Fig 3D**). VAT loss was significantly greater in patients with G2–3 tumors (−31.6 ± 11.2%) compared to those with G1 tumors (−8.8 ± 10.3%, p < 0.001; **Fig 3E**).

These findings indicate that patients with high-grade liposarcoma experience substantially greater losses of skeletal muscle and visceral fat during the disease course compared to those with low-grade tumors.

## Impact of treatment modalities on CT-morphometric body composition

To evaluate the impact of different treatment modalities on longitudinal changes in body composition, patients were stratified by therapeutic regimen: surgery only, surgery combined with chemotherapy, and surgery combined with radiotherapy. Patients undergoing radio- as well as chemotherapy (n = 3) were excluded due to low sample size.

Patients who underwent surgery and chemotherapy demonstrated the most pronounced reductions in SMI, with a mean decline of −44.3 ± 13.7%, significantly greater than in the surgery-only group (−22.1 ± 15.1%, p < 0.0001, **Fig 4A**). In contrast, patients receiving surgery and radiotherapy exhibited a comparable reduction in SMI (−21.3 ± 13.3%) to those treated with surgery alone (p = 0.44). Changes in PSMI followed a similar pattern, although differences between groups did not reach statistical significance. Patients treated with surgery only had a mean PSMI decline of −35.7 ± 16.4%, compared to −42.5 ± 15.1% in the chemotherapy group and −48.1 ± 10.0% in the radiotherapy group (**Fig 4B**). PMI decreased substantially and comparable across all treatment groups (surgery and radiotherapy: −74.1 ± 14.4%, surgery only: −57.0 ± 25.5% and surgery with chemotherapy: −57.2 ± 27.8%, **Fig 4C**). However, these differences did not reach statistical significance. SMD exhibited only modest and non-significant changes across all groups: −2.1 ± 24.5% in the surgery-only group, −5.1 ± 24.2% with chemotherapy, and −21.3 ± 13.3% following radiotherapy (**Fig 4D**). VAT showed a significantly greater reduction in patients treated with surgery and chemotherapy (−45.5 ± 10.1%) compared to the surgery-only group (−23.7 ± 15.2%, p < 0.0001, **Fig 4E**). Patients who received radiotherapy showed an intermediate VAT decline of −28.1 ± 5.1%, which was not statistically different from surgery only.

These findings suggest that the addition of chemotherapy to surgical treatment is associated with markedly greater losses in both skeletal muscle mass and visceral fat, while radiotherapy appears to exert a less pronounced impact on body composition.

## Impact of local tumor recurrence on CT-morphometric body composition

The presence of local tumor recurrence was associated with significantly more pronounced deterioration in multiple CT-based body composition parameters over the disease course.

Patients who experienced local recurrence demonstrated a significantly greater decline in SMI compared to those without recurrence (−36.3 ± 22.1% vs. −18.5 ± 22.6%, p < 0.01, **Fig 5A**). A similar pattern was observed for PSMI, with reductions of −49.2 ± 16.8% in the recurrence group versus −36.7 ± 18.8% in the non-recurrence group (p = 0.01, **Fig 5B**). PMI declined to a comparable extent in both groups (−58.0 ± 33.8% vs. −58.6 ± 20.9%, p = 0.33, **Fig 5C**).

SMD showed a marked difference between groups, increasing slightly in patients without recurrence (+2.1 ± 24.6%) but decreasing significantly in those with recurrence (−16.4 ± 17.8%, p < 0.01, **Fig 5D**). VAT loss was also significantly greater in the recurrence group (−34.4 ± 11.3%) compared to patients without local recurrence (−16.6 ± 13.3%, p < 0.001, **Fig 5E**).

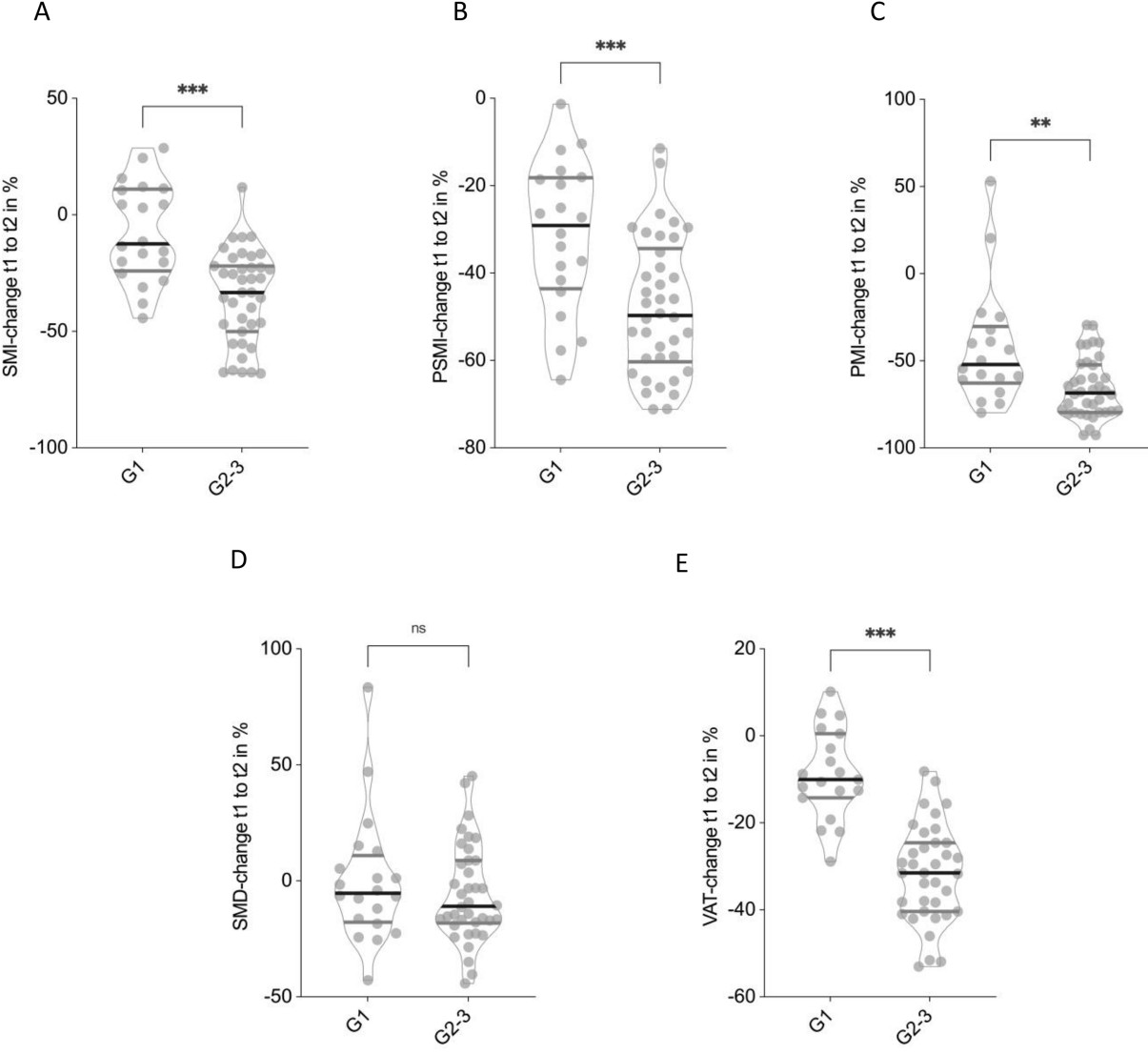

**Fig 3. Tumor grade-dependent differences in the percentage change of CT morphometric parameters.** Patients with grade 2-3 tumors showed significantly greater reductions in Skeletal muscle index (SMI) **(A)**, Paraspinal muscle index (PSMI) **(B)** and Psoas muscle index (PMI) **(C)**. No significant changes in Skeletal muscle index (SMD) between high- and low-grade tumors could be observed **(D)**, while Visceral adipose tissue declined significantly greater in the G2-3 tumor group **(E)**. Non-parametric Mann-Whitney U test has been applied.

These findings indicate that local tumor recurrence is associated with accelerated loss of skeletal muscle mass and visceral adipose tissue, as well as a significant decline in muscle quality.

### Effect of Baseline sarcopenia on survival and function

Baseline sarcopenia was defined using established sex-specific SMI cutoffs (<52.4 cm²/m² for males and <38.5 cm²/m² for females), originally derived in large cancer cohorts and subsequently widely applied in oncologic body composition research. Although these thresholds were not specifically developed in liposarcoma patients, they represent the most consistently used and externally validated CT-based cutoffs in cancer populations [24,25]. It was significantly associated with worse overall survival in liposarcoma patients (**Fig 6A**). Patients with sarcopenia at tCT1 had a median survival

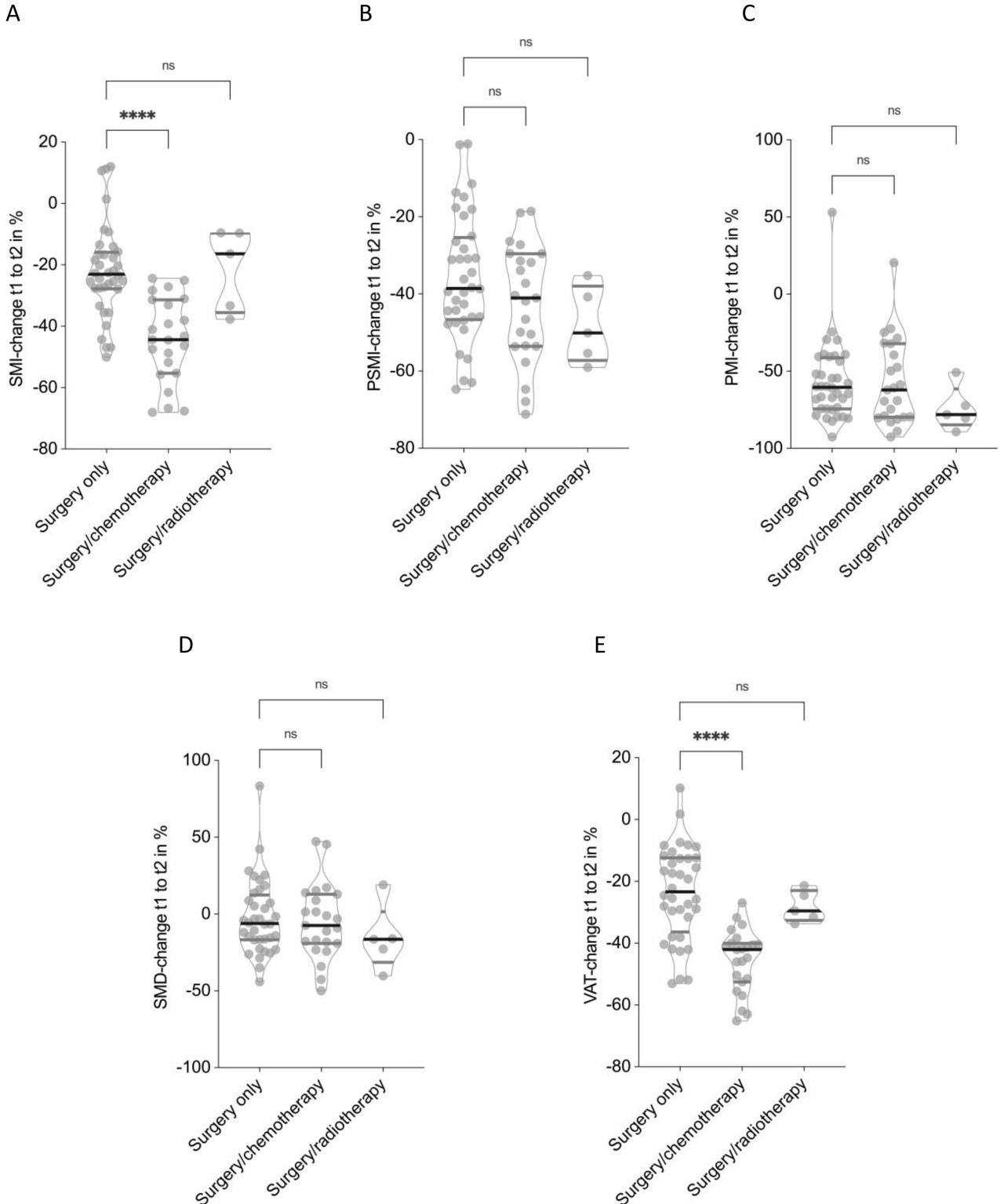

**Fig 4. Impact of treatment modality on body composition changes.** Chemotherapy was associated with significantly greater reductions in Skeletal muscle index (SMI) **(A)** and Visceral adipose tissue (VAT) **(E)** compared to surgery alone (p < 0.0001 and p < 0.0001, respectively), while radiotherapy did not significantly differ from surgery alone. Changes in Paraspinal muscle index (PSMI) **(B)**, Psoas muscle index (PMI) **(C)**, and Skeletal muscle density (SMD) **(D)** were not statistically significant. Non-parametric Kruskal-Wallis test has been applied.

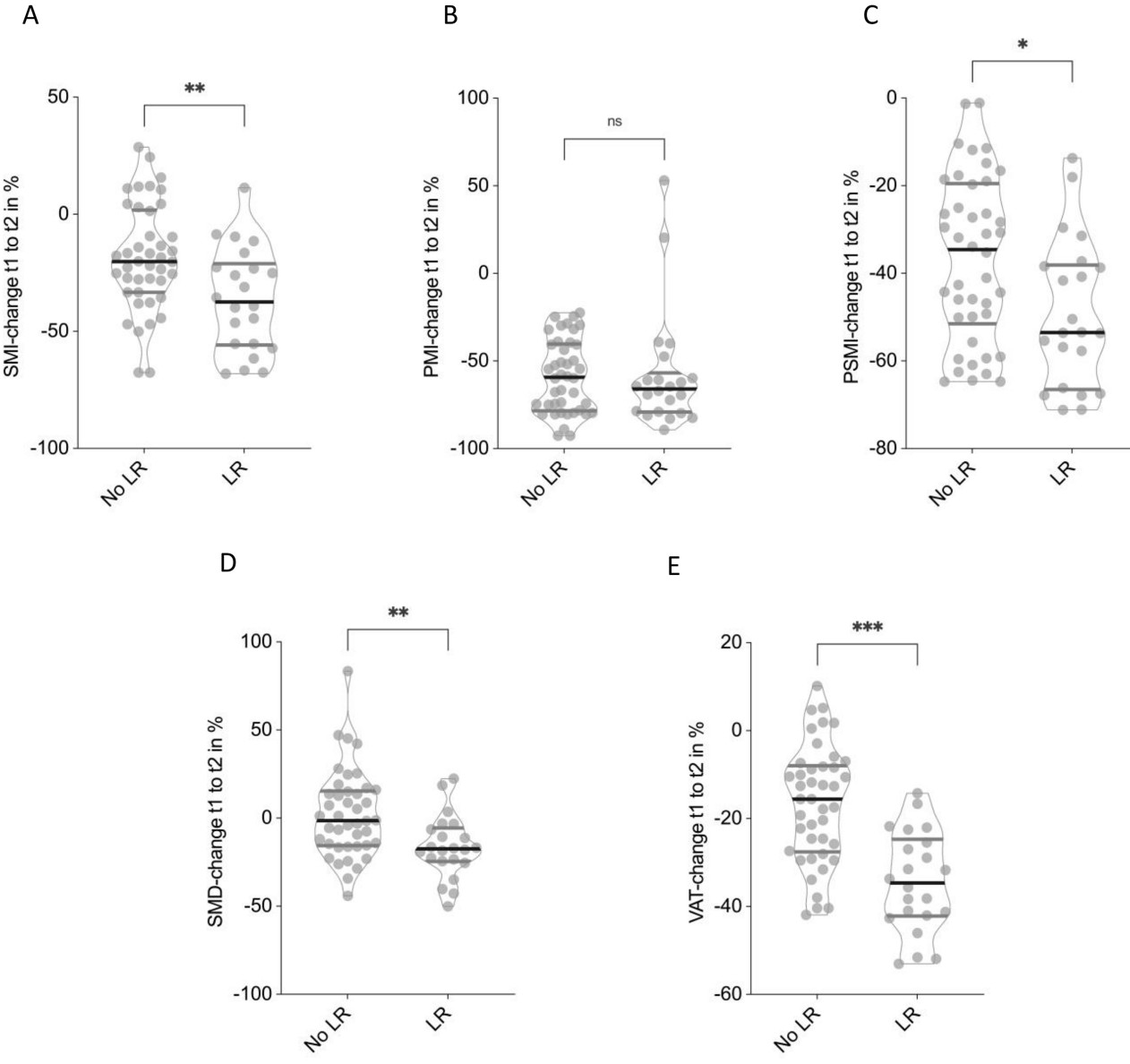

**Fig 5. Association between local tumor recurrence and changes in CT morphometric parameters.** Patients with local recurrence showed significantly greater declines in Skeletal muscle index (SMI) **(A)** and Paraspinal muscle index (PSMI) **(B)**. Psoas muscle index (PMI) showed no signficant differences **(C)**, while Skeletal muscle density (SMD) **(D)**, and Visceral adipose tissue (VAT) **(E)** did show significant differences between patients with and without local recurrence. Non-parametric Mann-Whitney U test has been applied.

of 38.0 months compared to 103.1 months in non-sarcopenic patients (p < 0.01). Despite the difference in survival, no clinically meaningful difference in functional status was observed between groups. ECOG performance status at follow-up remained stable in both cohorts (median ECOG = 1, range = 2) (**Fig 6B**).

### Impact of longitudinal SMI decrease on survival and function

Using a threshold determined by Youden's J from ROC curve analysis, a ≥ 15% decrease in SMI over the course of disease was identified as a critical cut-off for survival discrimination. Patients with a SMI decrease ≥ 15% experienced

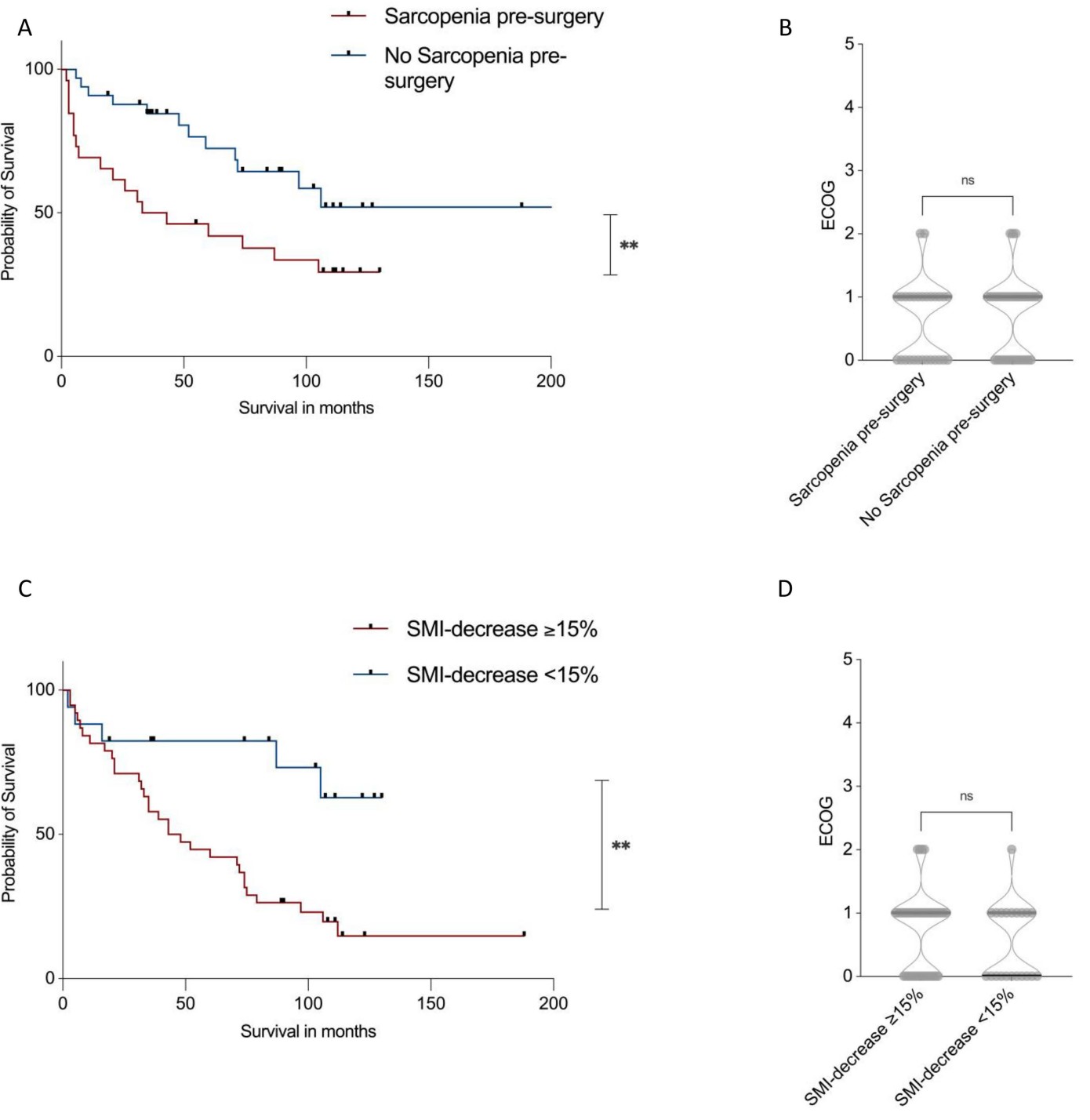

**Fig 6. Kaplan-Meier survival curve stratified by baseline sarcopenia.** Patients with pre-treatment sarcopenia had significantly shorter survival (p = 0.019, **A**), although no significant difference was observed in follow-up ECOG performance status **(B)**. Kaplan-Meier survival curve based on a ≥ 15% reduction in SMI during the disease course. Patients with ≥15% SMI loss had significantly reduced survival compared to those with <15% decline (p = 0.02, **C**), while follow-up ECOG performance status remained comparable (p = n.s., **D**). Log-rank test (Mantel-Cox test) was used for survival comparison, while non-parametric Mann-Whitney U test was applied for ECOG status comparison.

significantly reduced survival compared to those with less decline (**Fig 6C**). Median survival in the ≥ 15% loss group was 45.5 months, whereas the median survival for patients with <15% SMI decrease could not be calculated due to the high proportion patients that were still alive at follow-up (p < 0.01). As with baseline sarcopenia, no substantial difference in ECOG performance status was detected between groups at follow-up (median ECOG = 1, range = 2) (**Fig 6D**).

### Baseline sarcopenia and SMI decrease are independent predictors of survival

To determine whether both CT morphometric sarcopenia at baseline as well as sarcopenia progression over the disease could serve as independent predictors of poor survival, mono- and multivariate Cox Hazard Regression Models were used. In univariate Cox regression analysis, both baseline sarcopenia and a ≥ 15% SMI decrease were significantly associated with worse survival outcomes. Patients with baseline sarcopenia had a hazard ratio (HR) of 2.441 (95% CI: 1.517–4.111, p = 0.006), and those with ≥15% SMI loss had a HR of 2.597 (95% CI: 1.462–3.991, p = 0.005) (**Table 3**).

These associations remained significant in multivariate analysis after adjustment for age, sex, ECOG status, tumor grade, and local tumor recurrence. Baseline sarcopenia remained an independent predictor of worse survival with a HR of 2.331 (95% CI: 1.717–4.441, p = 0.007), and a ≥ 15% SMI decrease was likewise independently associated with mortality (HR: 2.601, 95% CI: 1.482–3.551, p = 0.006) (**Table 4**).

## Discussion

In this study, we provide comprehensive evidence that CT-morphometric parameters undergo significant deterioration over the disease course in liposarcoma patients and that both baseline sarcopenia and progressive skeletal muscle loss are

**Table 3. Univariate Cox proportional hazards regression analysis identifying predictors of overall survival. Both baseline sarcopenia and ≥15% SMI decline were significantly associated with reduced survival.**

| Predictor | Hazard Ratio | 95%-Confidence Interval | p-Value |
|---|---|---|---|
| Age | 0.979 | 0.951-1.008 | 0.147 |
| Sex | 2.046 | 0.892-4.693 | 0.091 |
| Local tumor recurrence | 1.331 | 1.112-3.331 | **0.028*** |
| ECOG | 1.188 | 0.550-2.569 | 0.661 |
| Tumor grade | 1.667 | 1.122-2.051 | **0.040*** |
| Baseline Sarcopenia | 2.441 | 1.517-4.111 | **0.006**** |
| ≥15% SMI decrease | 2.597 | 1.462-3.991 | **0.005**** |

**Table 4. Multivariate Cox proportional hazards regression analysis adjusted for age, sex, ECOG, tumor grade, and recurrence. Both baseline sarcopenia and ≥15% SMI decrease remained independent predictors of poorer survival.**

| Predictor | Hazard Ratio | 95%-Confidence Interval | p-Value |
|---|---|---|---|
| Age | 0.983 | 0.949-1-1.018 | 0.332 |
| Sex | 2.554 | 0.707-4.233 | 0.153 |
| Local tumor recurrence | 1.209 | 1.1661-3.001 | **0.018*** |
| ECOG | 1.739 | 0.685-3.414 | 0.245 |
| Tumor grade | 1.557 | 1.325-2.151 | **0.045*** |
| Baseline Sarcopenia | 2.331 | 1.717-4.441 | **0.007**** |
| ≥15% SMI decrease | 2.601 | 1.482-3.551 | **0.006**** |

independently associated with worse survival. These findings not only underscore the prognostic significance of sarcopenia in disease course of liposarcomas but also introduce CT-based body composition analysis as a meaningful, non-invasive tool for risk stratification.

Consistent with prior observations in gastrointestinal and thoracic malignancies, our data show that liposarcoma patients experience substantial declines in skeletal muscle indices (SMI, PSMI, PMI) and VAT over time, irrespective of sex [26–29]. Importantly, SMD, a surrogate for muscle quality, remained relatively stable, suggesting that the observed sarcopenia is predominantly quantitative rather than qualitative. The decline in body composition metrics was more pronounced in patients with high-grade tumors and local tumor recurrence indicating that tumor aggressiveness may accelerate muscle and fat loss through catabolic or inflammatory pathways [30–33].

This study also highlights the differential effects of systemic therapies on sarcopenia progression. Chemotherapy, but not radiotherapy, was associated with significantly accelerated declines in SMI and VAT, consistent with known cachectic effects of certain cytotoxic regimens [30–33]. This observation warrants further investigation, whether prehabilitation or muscle-preserving strategies could mitigate these adverse effects in selected patients [34].

Additionally, correlating these morphometric changes with clinical parameters such as tumor grade, recurrence, and treatment modality, we provide novel insights into the dynamic nature of body composition during disease progression and its potential as a monitoring tool.

Notably, both baseline sarcopenia and a ≥ 15% decrease in SMI during follow-up were independently predictive of reduced overall survival, even after adjustment for multiple other variables. This reinforces the hypothesis that morphometric deterioration reflects a distinct biological vulnerability not captured by conventional prognostic scores. Interestingly, despite their strong association with survival, neither baseline sarcopenia nor longitudinal SMI decline was reflected in functional deterioration as measured by ECOG performance status. This discrepancy may partially stem from the inherently subjective nature of the ECOG scale, which may not be sensitive enough to detect early or subclinical declines in physical function. Alternative tools, such as handgrip strength or gait speed [35], which are validated components of sarcopenia assessment, could provide more objective and nuanced evaluations of functional status in this context in future studies.

Our findings have important clinical implications. Given that cross-sectional imaging is already a standard component of oncologic surveillance, incorporating morphometric measurements into routine radiologic workflows could offer a low-cost, scalable method for identifying high-risk patients. Such assessments could be used to guide supportive interventions, including nutritional counseling, physical therapy, or early integration of palliative care services [36–38]. Furthermore, body composition trends could help clinicians anticipate treatment-related complications or poor recovery trajectories, particularly in patients considered for aggressive multimodal therapy.

Several limitations should be noted. First, the retrospective nature of the study introduces the possibility of selection bias, particularly regarding the availability of high-quality imaging and complete follow-up data. Second, while CT morphometry provides reliable estimates of muscle mass and adiposity, it does not capture functional muscle strength or physical performance directly. Third, the sarcopenia cutoffs applied in this study were adopted from prior work in mixed cancer populations and have not yet been validated specifically for liposarcoma. While these thresholds are widely used and prognostically relevant in several malignancies, variations in muscle mass distribution across tumor types and patient populations may limit direct transferability. To mitigate this, we additionally evaluated longitudinal SMI decline (≥15%), a within-patient trajectory measure independent of external cutoffs, which demonstrated robust prognostic value in multivariate models. Furthermore, CT-based morphometric measurements were performed by a single observer without formal inter-rater reliability assessment. However, the use of standardized HU thresholds and semi-automated segmentation reduces operator dependence and has been validated for reproducibility in prior work. Lastly, patients with local recurrence might be overrepresented, because this study was performed in a nationwide sarcoma center.

In summary, both baseline sarcopenia and progressive muscle loss over the disease course are strong, independent predictors of decreased overall survival in liposarcoma patients. These findings underscore the value of CT-based morphometric analysis as a clinically meaningful tool for monitoring disease trajectory and identifying vulnerable patients. Notably, patients with high grade tumors, undergoing chemotherapy appear to be particularly at risk for accelerated sarcopenia and fat loss, highlighting the importance of early risk stratification in this subgroup. Prophylactic interventions, including structured exercise programs and targeted nutritional supplementation, should be considered to mitigate treatment-related muscle wasting and preserve functional status. Future prospective studies are needed to validate these results and explore whether early interventions targeting muscle preservation can improve clinical outcomes in this patient population

## Supporting information

**S1 File. Revised final corrected liposarcoma study data.**
(XLSX)

## Author contributions

**Conceptualization:** Julian Kylies, Matthias Priemel, Anna Dupree, Karl-Heinz Frosch, Tobias M. Ballhause.

**Data curation:** Julian Kylies, Tobias M. Ballhause.

**Methodology:** Anna Dupree.

**Project administration:** Julian Kylies, Tobias M. Ballhause.

**Supervision:** Karl-Heinz Frosch.

**Validation:** Matthias Priemel.

**Writing – original draft:** Julian Kylies.

**Writing – review & editing:** Matthias Priemel, Anna Dupree, Karl-Heinz Frosch, Tobias M. Ballhause.

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
