## [Decision Letter · Decision Letter 0]

22 Oct 2025

Dear Dr. Kylies,

Thank you for submitting your manuscript to PLOS ONE. After careful consideration, we feel that it has merit but does not fully meet PLOS ONE’s publication criteria as it currently stands. Therefore, we invite you to submit a revised version of the manuscript that addresses the points raised during the review process.

We look forward to receiving your revised manuscript.

Kind regards,

Masaki Mogi

Academic Editor

PLOS ONE

Reviewers' comments:

Reviewer's Responses to Questions

**Comments to the Author**

1. Is the manuscript technically sound, and do the data support the conclusions?

Reviewer #1: Yes

2. Has the statistical analysis been performed appropriately and rigorously?

Reviewer #1: Yes

3. Have the authors made all data underlying the findings in their manuscript fully available?

Reviewer #1: Yes

4. Is the manuscript presented in an intelligible fashion and written in standard English?

Reviewer #1: Yes

Reviewer #1: Dear Authors,

Thank you for the opportunity to review your manuscript entitled “Sarcopenia as an Independent Prognostic Marker in Liposarcoma: A Longitudinal Analysis of Body Composition and Survival”

This is a well-designed and clinically relevant study addressing the prognostic role of sarcopenia in liposarcoma. However, the following are specific suggestions for revision:

- Please better describe the aims of the study.

- The cutoff values for sarcopenia are adopted from prior literature. However, are they validated specifically in cancer patients or sarcoma populations?

- Were the CT-based measurements conducted by a single observer, or was inter-rater agreement evaluated?

**Do you want your identity to be public for this peer review?** For information about this choice, including consent withdrawal, please see our Privacy Policy

Reviewer #1: No

---

## [Author Response · Author response to Decision Letter 1]

28 Oct 2025

Dear Prof. Mogi,

Dear Reviewer,

We sincerely thank you for the thorough and constructive review of our manuscript entitled "Sarcopenia as an Independent Prognostic Marker in Liposarcoma: A Longitudinal Analysis of Body Composition and Survival". We greatly appreciate the opportunity to revise our manuscript for PLOS One.

In the following point-by-point responses, we have addressed each of the reviewer's comments in detail. We have implemented revisions directly into the manuscript and indicated the respective changes.

We are confident that these revisions have significantly improved the clarity, scientific rigor, and clinical relevance of our work.

Reviewer #1

Reviewer comment: Dear Authors, Thank you for the opportunity to review your manuscript entitled “Sarcopenia as an Independent Prognostic Marker in Liposarcoma: A Longitudinal Analysis of Body Composition and Survival”.

This is a well-designed and clinically relevant study addressing the prognostic role of sarcopenia in liposarcoma. However, the following are specific suggestions for revision.

Response: We sincerely thank the reviewer for the thoughtful evaluation and for recognizing the clinical relevance and overall design of our study. We appreciate the constructive suggestions and have carefully revised the manuscript accordingly. In the following point-by-point responses, we address each comment in detail and highlight all corresponding changes made in the revised manuscript.

Reviewer comment: Please better describe the aims of the study.

Response: We have clarified the aims of the study in both Abstract and Introduction. Specifically, we clearly state that the study aims (1) to evaluate longitudinal changes in CT-morphometric body composition parameters throughout the disease course of liposarcoma, (2) to assess how tumor grade, local recurrence, and treatment modalities influence these morphometric trajectories and (3) to determine whether baseline sarcopenia and progressive muscle loss are associated with overall survival and functional outcomes.

"Abstract

Sarcopenia is increasingly recognized as an important prognostic factor in oncology; however, its clinical relevance in liposarcoma remains insufficiently defined. This study aimed to evaluate longitudinal changes in CT-derived body composition parameters in liposarcoma patients, to assess the influence of tumor grade, recurrence, and treatment modalities on these parameters and to determine the association of baseline sarcopenia and progressive muscle loss with overall survival and functional status." (P. 1, L. 26-31)

"Introduction

(...) Therefore, the aim of this study was to investigate body composition dynamics over the disease course in a well-characterized cohort of liposarcoma patients undergoing surgical resection. Specifically, we (1) evaluated longitudinal changes in CT-derived morphometric parameters (SMI, PSMI, PMI, SMD, and VAT), (2) examined the prognostic value of baseline sarcopenia and progressive skeletal muscle loss on overall survival and functional outcomes, and (3) analyzed the impact of tumor grade, recurrence, and treatment modalities on these body composition trajectories. By focusing exclusively on liposarcoma, this study provides entity-specific insights into the prognostic and clinical relevance of sarcopenia in soft-tissue sarcoma." (P. 2, L. 98-106)

Reviewer comment: The cutoff values for sarcopenia are adopted from prior literature. However, are they validated specifically in cancer patients or sarcoma populations?

Response: We thank the reviewer for this important comment. The SMI cutoff values applied in our study (52.4 cm²/m² for males and 38.5 cm²/m² for females) were originally established in a large cohort of patients with gastrointestinal and respiratory tract malignancies, where they were shown to be prognostic for survival. These cutoffs have subsequently been widely used and externally validated in several oncologic populations. However, it is correct that they have not been exclusively validated in sarcoma cohorts so far, and population-specific thresholds for this tumor entities are not yet available.

To address this limitation, we have now clarified the rationale for selecting these cutoffs in the Results section and have explicitly acknowledged the lack of liposarcoma-specific validation in the Discussion. Additionally, we now emphasize that our analysis also examined relative SMI loss (≥15% during disease course), which represents an internal, change-based measure not dependent on external reference equations and demonstrated independent prognostic value.

"Results

(...) Baseline sarcopenia was defined using established sex-specific SMI cutoffs (<52.4 cm²/m² for males and <38.5 cm²/m² for females), originally derived in large cancer cohorts and subsequently widely applied in oncologic body composition research. Although these thresholds were not specifically developed in liposarcoma patients, they represent the most consistently used and externally validated CT-based cutoffs in cancer populations." (P. 7, L. 350-354)

"Discussion - Limitations

(...) Third, the sarcopenia cutoffs applied in this study were adopted from prior work in mixed cancer populations and have not yet been validated specifically for liposarcoma. While these thresholds are widely used and prognostically relevant in several malignancies, variations in muscle mass distribution across tumor types and patient populations may limit direct transferability. To mitigate this, we additionally evaluated longitudinal SMI decline (≥15%), a within-patient trajectory measure independent of external cutoffs, which demonstrated robust prognostic value in multivariate models." (P. 9, L. 459-465)

Reviewer comment: Were the CT-based measurements conducted by a single observer, or was inter-rater agreement evaluated?

Response: We thank the reviewer for this relevant clarification request. All CT-based morphometric measurements in this study were performed by a single observer trained in quantitative body composition analysis. To ensure consistency and reproducibility, we employed a semi-automated segmentation workflow using standardized Hounsfield Unit thresholds and fixed anatomical landmarks at L3. This approach minimizes subjective variation and has been widely used in oncologic body composition research. We have now clarified this in the Methods section. Although inter-rater agreement was not formally assessed in this retrospective study, the semi-automated standardized measurement procedure reduces operator dependency. We now acknowledge this as a study limitation in the Discussion.

"Materials and Methods

(...) All CT-based morphometric measurements were performed by a single trained observer using a semi-automated segmentation workflow to ensure consistency across scans. Standardized Hounsfield Unit thresholds and pre-defined region-of-interest boundaries at the L3 vertebral level were applied to minimize operator-related variability. Formal inter-rater reliability testing was not conducted in this retrospective dataset." (P. 3, L. 144-148)

"Discussion

(...) Furthermore, CT-based morphometric measurements were performed by a single observer without formal inter-rater reliability assessment. However, the use of standardized HU thresholds and semi-automated segmentation reduces operator dependence and has been validated for reproducibility in prior work." (P. 10, L. 465-469)

We again thank the reviewer and the editorial team for their valuable feedback and guidance.

We hope that the revised manuscript now meets the expectations for PLOS One and remains suitable for publication. We are grateful for the opportunity and look forward to your feedback.

Sincerely,

Dr. Julian Kylies, MD

Sarcoma Center Hamburg,

University Medical Center Hamburg-Eppendorf

---

## [Editor Report · Decision Letter 1]

30 Nov 2025

Sarcopenia as an Independent Prognostic Marker in Liposarcoma: A Longitudinal Analysis of Body Composition and Survival

PONE-D-25-31708R1

Dear Dr. Kylies,

We’re pleased to inform you that your manuscript has been judged scientifically suitable for publication and will be formally accepted for publication once it meets all outstanding technical requirements.

Kind regards,

Masaki Mogi

Academic Editor

PLOS ONE

Additional Editor Comments (optional):

The authors have well responded to the Reviewer’s suggestions.
---

## [Editor Report · Acceptance letter]

PONE-D-25-31708R1

PLOS One

Dear Dr. Kylies,

I'm pleased to inform you that your manuscript has been deemed suitable for publication in PLOS One. Congratulations! Your manuscript is now being handed over to our production team.

Kind regards,

on behalf of

Dr. Masaki Mogi

Academic Editor

PLOS One